# Biomechanical Evaluation of Hydroxyapatite/poly-l-lactide Fixation in Mandibular Body Reconstruction with Fibula Free Flap: A Finite Element Analysis Incorporating Material Properties and Masticatory Function Evaluation

**DOI:** 10.3390/bioengineering11101009

**Published:** 2024-10-11

**Authors:** Sang-Min Lee, Tae-Gon Jung, Won-Hyeon Kim, Bongju Kim, Jee-Ho Lee

**Affiliations:** 1Department of Oral and Maxillofacial Surgery, Asan Medical Center, College of Medicine, University of Ulsan, 88 Olympic-ro, Songpa-gu, Seoul 05505, Republic of Korea; smlee900@gmail.com; 2Medical Device Development Center, Osong Medical Innovation Foundation, Chungju 28160, Republic of Korea; bygon@kbiohealth.kr; 3Implant R&D Center, OSSTEM IMPLANT Co., Ltd., Seoul 07789, Republic of Korea; wonhyun79@gmail.com; 4Dental Life Science Research Institute, Seoul National University Dental Hospital, Seoul 03080, Republic of Korea

**Keywords:** reconstruction surgery, fibula free flap, finite element analysis, HA-PLLA, titanium, fixation system

## Abstract

In reconstructive surgery following partial mandibulectomy, the biomechanical integrity of the fibula free flap applied to the remaining mandibular region directly influences the prognosis of the surgery. The purpose of this study is to evaluate the biomechanical integrity of two fixation materials [titanium (Ti) and hydroxyapatite/poly-L-lactide (HA-PLLA)]. In this study, we simulated the mechanical properties of miniplate and screw fixations in two different systems by finite element analysis. A three-dimensional mandibular model was constructed and a fibula free flap and reconstruction surface were designed. The anterior and posterior end of the free flap was positioned with two miniplates and two additional miniplates were applied to the angled area of the fibula. The masticatory loading was applied considering seven principal muscles. The peak von Mises stress (PVMS) distribution, size of fixation deformation, principal stresses on bones, and gap opening size were measured to evaluate the material properties of the fixation. In the evaluation of properties, superior results were observed with both fixation methods immediately after surgery. However, after the formation of callus between bone segments at 2 months, the performance of Ti fixation decreased over time and the differences between the two fixations became minimal by 6 months after surgery. The result of the study implies the positive clinical potential of the HA-PLLA fixation system applied in fibula free flap reconstruction.

## 1. Introduction

Reconstructive surgery in the oral and maxillofacial region is essential for restoring a patient’s facial appearance, masticatory function, and speech. In cases of extensive mandibular defects due to diseases such as oral cancer and medication-related osteonecrosis, the reconstruction can be performed by using free flaps from the patient’s own body. Through microvascular surgery, an autograft of soft tissue, bones, artery, and vein can be transplanted to the defect area of the mandible [1]. Well-known donor sites of such free flaps are the iliac crest, fibula, and scapula [2,3]. Among various types of osteocutaneous free flaps, the fibula free flap is currently regarded as the standard treatment for segmental mandibular reconstruction [4]. Also, according to a previous study, the fibula free flap was the most preferred donor site when extensive mandible defect reconstruction was necessary [3]. A fibula free flap is an excellent option for reconstructing large mandibular defects. It offers a substantial amount of bone, often exceeding 20 cm, and results in minimal functional and aesthetic issues at the donor site. Also, oral rehabilitation with dental implant treatment is possible and the success rate of implants placed within the fibula free flap is excellent [5,6].

In reconstructive surgery, the optimal choice of fixation system can have a significant impact on the surgical outcome. Especially in the reconstruction of mandible defects, a reconstruction plate has been the most commonly used material in conventional fixation methods. This preference is due to its advantages including using a single plate, fewer screws, load-bearing capability, quick restoration of oral function, and the ability to adapt to the mandible’s curvature. This results in an enhanced flap–mandible contact after the adaptation of the fibula free flap [7]. However, there are several disadvantages such as plate exposure, malunion or nonunion of the neomandible, difficulty in intraoperative manipulation, interference with the vascular pedicle, and a large profile, which increases the risk of postoperative infection, potentially leading to osteonecrosis of the neomandible [8,9]. Several studies on mandibular reconstruction using miniplates or reconstruction plates with free flaps have been compared. According to Zhang et al., the results of a meta-analysis showed that miniplates and reconstruction plates are both appropriate for mandibular reconstruction using a vascularized osteocutaneous flap [8]. The research evaluated plate exposure, plate fracture/removal, infection, and overall complications and no significant difference was found between groups that used the reconstruction plate or miniplate. Also, Robey et al. specifically compared the use of reconstruction plates and miniplates in cases of mandibular reconstruction using a fibula free flap. Their findings reported no significant differences between the two methods [9]. This suggests that miniplates can be useful alternatives to reconstruction plates in mandibular reconstruction surgeries.

Meanwhile, fixation systems can be classified by the material. Conventionally, Ti fixation systems have been regarded as the gold standard for internal fixations in open reduction due to their strength, rigid characteristics, biocompatibility, and handling properties [10,11]. However, Ti fixation systems have disadvantages such as sensitivity to temperature [12], tactile sensation of plates and screws [13], potential growth restrictions [14], and interference with imaging and radiotherapy [15]. As an alternative to Ti fixation systems, biodegradable materials such as hydroxyapatite poly-L-lactide (HA-PLLA) (Osteotrans-MX; Takiron Co., Umeda, Japan) have been introduced to reconstructive surgery due to the possibility of such materials overcoming the disadvantages of Ti fixation systems [10]. Research on biodegradable fixation systems has been continuously conducted, and there have been reports on the physical properties and clinical efficacy of these materials. Previous studies reported a clinical evaluation of u-HA/PLLA composite devices used for the internal fixation of mandibular fractures in patients and the system demonstrated adequate mechanical strength for providing rigid fracture fixation [16,17]. Also, when compared to the Ti fixation system, biodegradable fixation showed equivalent stability [18]. Moreover, an in vitro study was conducted that compared the mechanical properties of Ti and biodegradable fixation systems under loading and the results suggested that there was no significant difference between the two systems [19]. Several studies have evaluated the mechanical properties of internal fixation devices made from different materials for facial trauma using finite element analysis (FEA). Park et al. and Jung et al. simulated the mechanical properties of fixation systems made from titanium, magnesium, poly-L-lactide, and HA-PLLA applied for facial traumas [10,20]. Jung et al. compared the biomechanical integrity of Ti and HA-PLLA fixation systems used in reconstruction surgery with a deep circumflex iliac artery (DCIA) free flap through FEA simulation [21]. The findings from previous studies consistently indicate that HA-PLLA fixations could serve as a viable alternative to titanium fixations. In addition, HA-PLLA is the most advanced biodegradable material currently available. Traditional absorbable materials, which used PLLA alone, had drawbacks such as low strength, poor osseointegration, and a long absorption time. However, HA-PLLA overcame these limitations by incorporating hydroxyapatite [22,23,24]. Therefore, this study has selected HA-PLLA as the simulation target due to its potential as a biodegradable material. The aim of this study was to estimate and analyze the mechanical properties and biocompatibility of two different materials applied in fibula free flap reconstruction after segmental mandibulectomy. The biomechanical integrity of Ti and HA-PLLA fixation systems were compared using FEA simulations.

## 2. Materials and Methods

### 2.1. FEA Model

This retrospective study was approved by the Institutional Review Board (IRB number 2024-0635) of Asan Medical Center. FEA models of the mandible were generated using a cone-beam computed tomography (CBCT) image of a patient suffering from severe chronic osteomyelitis of the mandible. The patient had undergone mandibular resection, and the subsequent reconstruction involved the use of a fibula free flap.

An FEA model was first constructed based on the CBCT data, prioritizing the representation of clear structures such as bone and teeth. Subsequently, within the bone morphology, cortical bone was assumed to have a thickness of 2 mm, while cancellous bone was treated separately. In the case of the periodontal ligament (PDL), which cannot be distinguished in CBCT images, this was implemented with a thickness of 0.2 mm between the mandible and teeth [25,26]. In such instances, the rationale for these assumptions was established by referencing prior studies. Then, healthy control mechanical properties were applied. The FEA model is composed of a tetrahedral structure consisting of nodes and elements, and the accuracy of FEA increases with a greater number of nodes and elements.

Bujtar et al. utilized Hounsfield units (HU) obtained from CBCT images and developed equations to calculate the apparent density and elastic modulus (Equations (1) and (2)) [27]. The elastic moduli of teeth and cortical and cancellous bones were determined using ρ_app_ (in kg/m^3^) and Young’s modulus E (in MPa). The fibular bone and callus properties were assessed by measuring Hounsfield units from CBCT images over time. Subsequently, Young’s modulus was applied using the equation. To ensure consistency, the material properties were referenced from previous studies [27,28,29,30,31,32,33].

The FEA simulations considered the material properties of cortical bone and callus in a fibula free flap at three distinct time intervals: 2 weeks, 2 months, and 6 months post-surgery. The material property values of the components were modified at each time point (Table 1).
ρ_app_ = −200 + 1.2 × HU(1)
E = 0.024 × ρ_app_^1.762^(2)

Two different types of miniplates were applied in the simulation. The four-hole miniplate that was designed was 24.2 mm in length, 4.2 mm in width, and 1 mm in thickness. The dimensions of the six-hole miniplate were identical to those of the four-hole miniplate, except for its length, which measured 34.2 mm (Figure 1a). The screw head and body had diameters of 2.3 mm and 2 mm, respectively, with a screw length of 6 mm (Figure 1b). This study evaluated the mechanical properties of screws and miniplates, fabricated from either Ti or HA-PLLA. The two fixations were the same size.

FEA models were converted into meshes using specialized mesh generation software (Altair HyperWorks v17.0; Altair Engineering Inc., Troy, MI, USA). The cortical and cancellous bones, tooth, PDL, fibular bone, callus miniplates, and screws were implemented with a linear tetrahedral mesh (Table 2). The authors of [34] carried out FEA with implants and derived an optimal mesh size of 0.3 mm through mesh convergence. In our study, we set the maximum mesh size to 0.3 mm for implants, PDL, and teeth, which is smaller than the optimal mesh size in [34]. However, our results indicated that cancellous bone was not a significant factor due to its minimal contribution to the mechanical integrity of the fibula free flap. As a result, cancellous bone was not included in our FEA model interpretation.

Using FEA software (ABAQUS CAE2016; Dassault Systèmes), a simulated fibula free flap was employed to replace the resected section of the mandible, extending from the midline to a segment just before the left ascending ramus of the mandible. Two weeks after the surgery, the fibula free flap model was configured to depict a non-union state, incorporating a gap between the residual mandible’s cortical bones and the fibula free flap (Figure 1c). The FEA models considered bone healing and formation over the 2- and 6-month periods after surgery, requiring adjustments to the material properties to represent a union state (Table 1 and Figure 1c).

### 2.2. Evaluation of Masticatory Function and Occlusal Force in FEA Model

One 4-hole miniplate and one 6-hole miniplate were applied at the midline region of the mandible, connecting it to the free flap. Additionally, two 4-hole miniplates were used at the juncture where the fibula free flap bends to align with the appropriate contour of the mandible body. Two 4-hole miniplates were used to connect the residual portion of the posterior mandible to the free flap. The simulation included screw holes for all six plates, allowing for the installation of up to 26 screws. However, 3 of the 26 screws were excluded from the FEA simulation to prevent potential invasion of the resection border during thread engagement.

In line with the previous studies conducted by Park et al. and Jung et al. [10,20], tie contact conditions were applied to the interfaces between screws and bones, miniplates and screws, and cortical and cancellous bones. The tie contact condition assumed complete unity at the interfaces between cortical and cancellous bones or full integration of the bone and implant [20,35,36]. The contact points between the bone portion of the free flap and the miniplates were considered to be in a sliding condition, with a friction coefficient of 0.5.

Masticatory function and movement of the mandible rely on several muscles such as the masseter muscle, pterygoid muscle, and temporalis muscle. Therefore, after reconstruction surgery, the bone and fixations in the reconstructed area gradually receive the force of muscles over time. Consequently, this leads to differences in occlusal patterns and occlusal forces.

Various parameters, including peak von Mises stress (PVMS) distributions, fixation deformations, principal stresses on cortical bones, and the gap opening distance, were measured. To assess the biomechanical effects of fixations on the fibula free flap, this study simulated static clenching tasks.

Loai Hijazi et al. evaluated stress distribution on the mandible and condylar fracture osteosynthesis during six types of clenching tasks accounting for seven principal muscles [37]. A similar type of clenching task was simulated by Wei Zhou et al., who used four pairs of vectors representing masticatory muscles to simulate the appropriate loading of the clenching task [38]. We adopted these two conditions to our simulation. However, two types of clenching tasks were eliminated in the model interpretation due to the absence of teeth on the left side of the FEA reconstruction model. Based on the previous literature, various occlusal conditions were considered. Since each occlusal condition represents a static state, the mandibular condyle was fully fixed in all directions for the simulation. Depending on the clenching condition, the teeth were also fully fixed in all directions. For the INC motion, only the anterior teeth were constrained, while for the ICP motion, only the canines and premolars were constrained. Additionally, the RMOL motion constrained only the molars while the RGF motion constrained the canines, premolars, and molars (Figure 2). The raw data for these loading conditions were sourced from a study conducted by Korioth and Hannam (1994) [39].

## 3. Results

### 3.1. PVMS Distributions and Patterns

The PVMS distribution data of the materials collected by clenching task simulations at three different time points are shown in Table 3. Two weeks after the surgery, the PVMS for the Ti screws ranged from 136.7 to 693.6 MPa, and for the Ti miniplates, it ranged from 188.7 to 1005 MPa. In the period from 2 months to 6 months postoperation, the PVMS for the Ti screws varied from 85.76 to 379.3 MPa, while for the Ti miniplates, it ranged from 103.0 to 549.9 MPa. In the case of the HA-PLLA screws, the measured PVMS values 2 weeks after the surgery ranged from 64.2 to 198.7 MPa while those of the HA-PLLA miniplates ranged from 71.6 to 268.0 MPa. In the period from 2 months to 6 months post operation, the PVMS of the HA-PLLA screws ranged from 22.7 to 84.44 MPa and that of miniplates ranged from 26.4 to 112.8 MPa. Figure 3 illustrates the PVMS distribution across the two fixation systems at three different follow-up time points.

When the principal stress was measured over time in healthy control cortical bone and fibula free flap bone (Figure 4), significant changes in principal stress were observed between 2 weeks and 2 months after surgery for both Ti and HA-PLLA in clenching tasks, stabilizing after that time. INC had the smallest stress change, increasing in the order of RMOL, RGF, and ICP. In fibula free flap bone, significant stress changes continued even after 2 months, unlike the stable trend in healthy bone.

### 3.2. Fixation System’s Deformation with Clenching Tasks

For Ti fixation, the range of deformation values at 2 weeks after surgery was between 0.239 and 1.065, while for HA-PLLA, it ranged from 0.301 to 1.353. Deformations greater than 1.00 mm were observed only in the measurements taken 2 weeks after surgery for both Ti and HA-PLLA, specifically in the ICP clenching task. Both screws and miniplates showed a relatively significant deformation tendency during the ICP clenching task. Regardless of the material type or clenching task, there was a trend of decreasing in deformation over time after surgery, and at the 6-month postoperative point, deformations were observed to be less than 0.85 mm for all fixations (Table 4 and Figure 5).

## 4. Discussion

A single plate (a reconstruction plate) is often applied for the fixation of fibula free flaps [40]. However, the choice between using miniplates or single plates (reconstruction plates) for fibula free flap fixation is selective and depends on clinical experience and the surgeon’s preference, with reports indicating that there is little difference in prognosis and outcomes between the two methods [41]. In this study, we adopted a method using multiple miniplates for the fixation of the fibula free flap. This approach was chosen to achieve a more stable load distribution and fixation strength by attaching fixation devices to both the upper and lower portions of the bone in situations where there are two or more osteotomy sites.

The mechanical integrity of a reconstructed mandible using a fibula free flap mainly relies on the secure placement of the fixation plates and screws on the cortical bones. Cortical bones possess a consistent and dense structure, providing a reliable base for securing screws and miniplates. Conversely, cancellous bones consist of spongy material, with trabeculae forming an irregular arrangement of thin columns and spaces. Therefore, cancellous bones are less effective at enduring masticatory forces than cortical bones. (Table 1) [21]. As a result, the majority of stress on mandible bones is directed towards the cortical bones. According to previous studies, simplification of the model is crucial for improving the accuracy of results in simulations of complex FEA models [42]. The contribution of cancellous bones to the mechanical integrity of the fibula free flap is minimal, and their irregular structure could lead to uncertainty and variability in finite element analysis. As a result, the interpretation of cancellous bones was excluded from the FEA simulation results. To maintain consistency, the mechanical properties of fixation materials, healthy control bones, Young’s modulus, and Poisson’s ratios were obtained from the previous literature, while the properties of teeth and the PDL were also incorporated based on findings from another study [10,20,43,44].

In previous FEA studies, the mechanical properties of the fixation system were evaluated under simple occlusal loading conditions. The load was incrementally increased at three measurement points until it reached a predetermined maximum load. This method was used to measure parameters such as the PVMS, the degree of deformation of the fixation system, and the principal stress applied to the bone [21]. However, in this study, we conducted FEA simulations using masticatory forces from seven principal muscles during six clenching tasks. Due to the reconstruction of the left side of the mandible with a fibula free flap and the absence of teeth, the left unilateral molar clench (LMOL) and left group function (LGF) tasks were excluded. This approach advances previous research by offering a more practical and clinically relevant scenario compared to simple occlusal loading conditions.

During clenching tasks, the Ti fixation system maintained consistent maximum and minimum tensile stress on the healthy control cortical bone over 2 weeks, 2 months, and 6 months postoperation. In contrast, the HA-PLLA fixation system showed greater variability in tensile stress levels across these time points (Figure 4a). The HA-PLLA fixation system showed the highest tensile stress in the order of ICP, RMOL, INC, and RGF, with ICP being 1.5 times higher at 2 weeks postoperation compared to 2 and 6 months. The minimum tensile stress followed the same order. The largest stress difference for ICP was between 2 weeks and the later time points. The fibula free flap had substantial fluctuations in principal stresses over time. Overall, the HA-PLLA system exhibited more notable stress variations than the Ti system. (Figure 4b). Both fixation systems showed decreased tensile stress over 2 weeks, 2 months, and 6 months postoperation, with values lower than those of healthy cortical bone. The most significant changes were observed in ICP measurements. As new bone fills the space between the resected mandible and the fibula free flap, tensile stress decreases, indicating stress transfer to stabilized cortical bones during healing. Bone healing reduces biomechanical stress on the remaining bones and fixations. Increasing HUs and densities suggest improved bone density as healing progresses.

Two weeks after the reconstruction surgery, the rebuilt mandible was not sufficiently robust to withstand masticatory forces, and the PVMS was focused on the fixations rather than being evenly distributed across the bones. The titanium fixation systems consistently bore greater stress than the HA-PLLA systems throughout the study period (Table 3). The Ti fixation systems showed higher PVMS and maintained strength and stiffness better than HA-PLLA fixations at 2 weeks and 2 months postoperation. Both fixation types experienced a significant decrease in PVMS by 2 and 6 months postoperation, with Ti showing a more marked decline. This decrease is likely due to the progressive integration of new bone, which improves stress distribution. If strength and stiffness are well-maintained up to the 2-month mark, HA-PLLA fixation can be considered comparable to Ti fixation in terms of mechanical performance.

Unlike the findings of previous studies by Park et al. and Jung et al., the screws in our fibula free flap simulation exhibited higher PVMS and deformation compared to those in the miniplates [10]. Specifically, the titanium screws had the highest PVMS concentration immediately after the surgery. The HA-PLLA plate showed about 1.5 times higher PVMS values than its screws, while Ti fixation showed no significant difference between the plate and screws. HA-PLLA fixation had PVMS values over three times higher than those of Ti fixation, indicating greater deformation, especially immediately after surgery. By 2 months postoperation, HA-PLLA fixations showed reduced deformation and more uniform PVMS distribution due to screw integration with bone. During clenching, PVMS values were highest in the ICP condition, likely due to the greatest occlusal force.

For both fixations, simulation of clenching tasks revealed that deformation was highest at 2 weeks postoperation and declined over time (Figure 5). When analyzing the size of deformity according to the clenching task, both fixation materials showed the greatest deformity in the order of ICP, RMOL, INC, and RGF. The deformation was particularly remarkable during ICP and RMOL. Quantitative analysis revealed that the deformation during ICP and RMOL was more than twice the size of the deformation observed during INC and RGF. This is likely because, during tasks that apply actual muscle force, more muscles and greater force are involved in ICP and RMOL, which include the masticatory force of the posterior teeth, compared to the forces involved in cutting with the anterior teeth or lateral movement.

Initially, the Ti fixation systems showed higher PVMS and better performance with principal stress, indicating superior mechanical strength and stiffness immediately after surgery compared to the HA-PLLA systems. However, the HA-PLLA fixations exhibited adequate mechanical integrity, with the PVMS and deformation values decreasing significantly over time, suggesting that these materials can become more integrated with the bone as healing progresses. By 2 months postoperation, both fixation systems displayed similar stress distributions and mechanical stability, indicating that HA-PLLA can be a suitable alternative to Ti fixations, particularly considering its biocompatibility and potential to overcome the disadvantages associated with Ti materials, such as sensitivity to temperature and interference with imaging. From 2 months postoperation, the biomechanical differences between the HA-PLLA and Ti fixations become negligible due to new bone formation. Therefore, when HA-PLLA fixation is applied, it is very important to stabilize the patient’s occlusion and minimize mobilization caused by mastication during the first 2 months. For example, intermaxillary fixation using elastics can be applied, and the patient should be restricted to a soft foods diet. The FEA simulations, incorporating realistic masticatory forces from principal muscles during various clenching tasks, provided a comprehensive assessment of the biomechanical performance of these fixation systems in a clinically relevant context. In terms of postoperative oral rehabilitation, the advantage of the HA-PLLA system became more pronounced. The titanium fixation system may need removal if exposed intraorally or extraorally, if inflammation occurs, or if it interferes with flap implantation or dental implant treatment. In contrast, the HA-PLLA system, which fuses with bone, rarely requires removal. Thus, for restoring occlusion with implants on a fibula free flap, the HA-PLLA system is likely more effective than the titanium system.

## 5. Conclusions

This study used FEA simulation to compare the mechanical properties and biocompatibility of Ti and HA-PLLA in fibula free flap reconstruction after segmental mandibulectomy. The findings highlight HA-PLLA’s potential as a promising fixation material, offering similar biomechanical benefits to Ti with added biocompatibility and patient comfort. In addition, the results show that for HA-PLLA to become a viable alternative to Ti fixation, the most important factor is to control the patient’s masticatory force and ensure occlusal stability within the first 2 months after surgery. Future research should explore long-term clinical outcomes and broader applications of HA-PLLA in reconstructive surgeries.

## Figures and Tables

**Figure 1 bioengineering-11-01009-f001:**
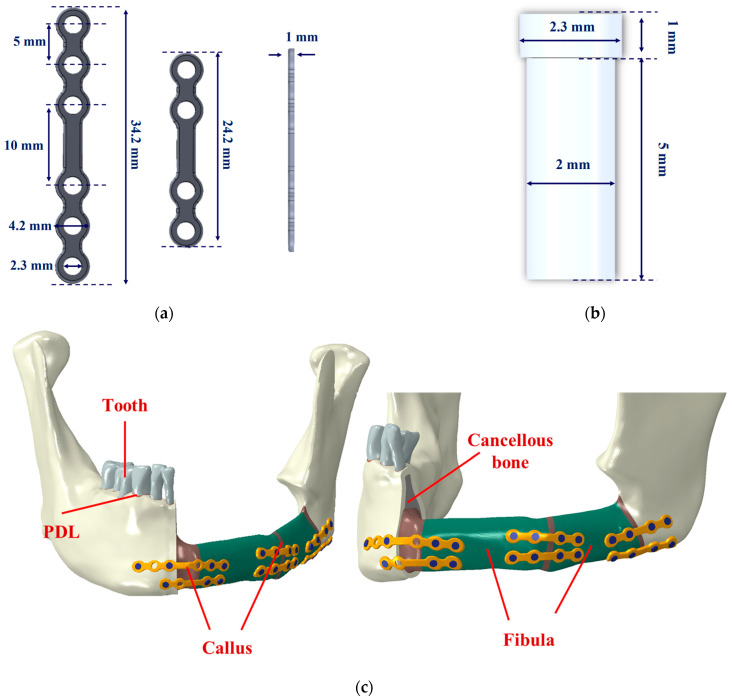
(**a**) Miniplate, (**b**) screw, and (**c**) mandibular reconstruction model after surgery. Screws and miniplates were implanted at the mandibular angle and chin surfaces.

**Figure 2 bioengineering-11-01009-f002:**
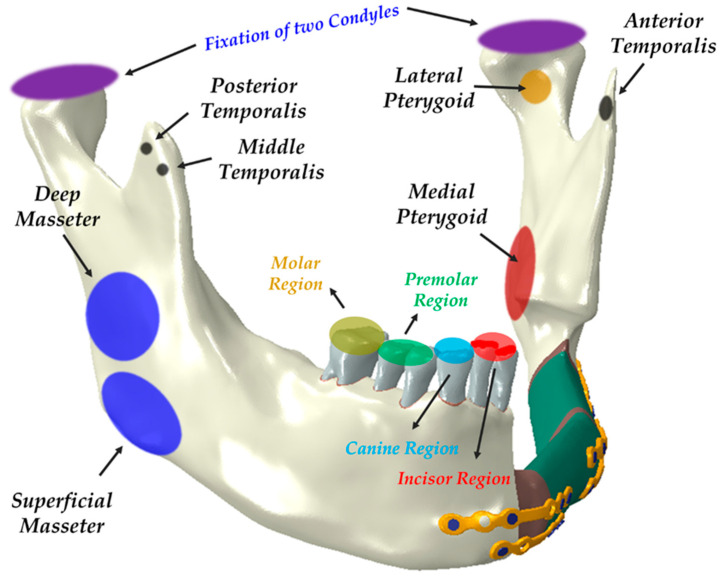
Simulated masticatory loading accounting for muscle force and boundary conditions on a reconstructed mandible. The image shows the seven principal muscles: superficial masseter (SM), deep masseter (DM), medial pterygoid (MP), lateral pterygoid (LP), anterior temporalis (AT), middle temporalis (MT), and posterior temporalis (PT).

**Figure 3 bioengineering-11-01009-f003:**
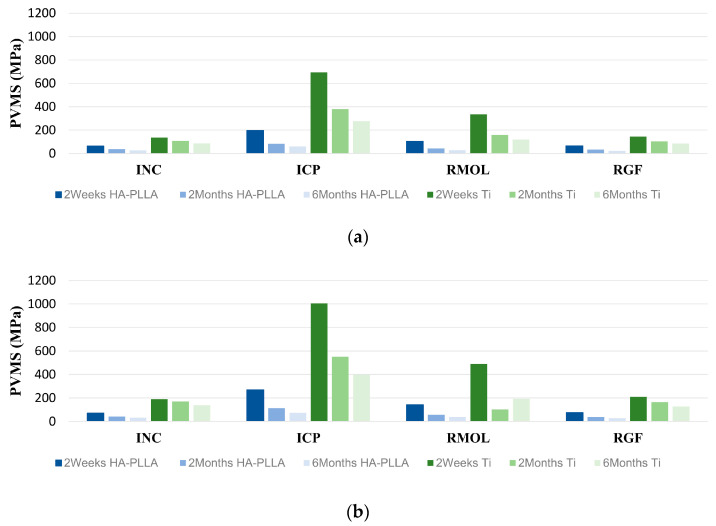
PVMS distribution trends across fixation systems at 2 weeks, 2 months, and 6 months after surgery. (**a**) Screws and (**b**) miniplates. Four types of clenching tasks: incisal clench (INC), intercuspal position (ICP), right unilateral molar clench (RMOL), and right group function (RGF).

**Figure 4 bioengineering-11-01009-f004:**
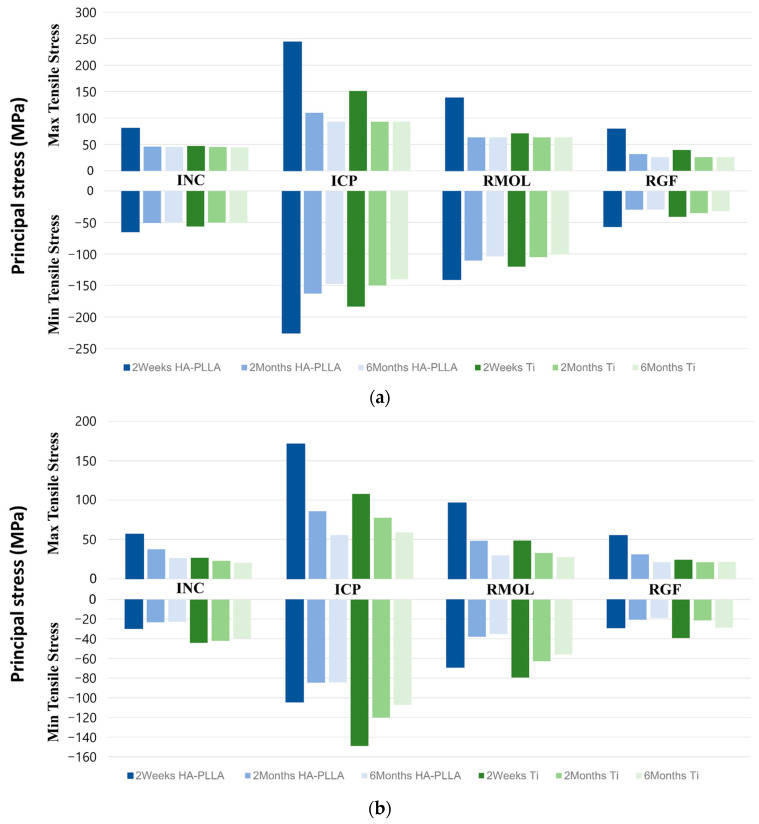
Change in the principal stresses. (**a**) Healthy control cortical bones and (**b**) fibular free flap bones.

**Figure 5 bioengineering-11-01009-f005:**
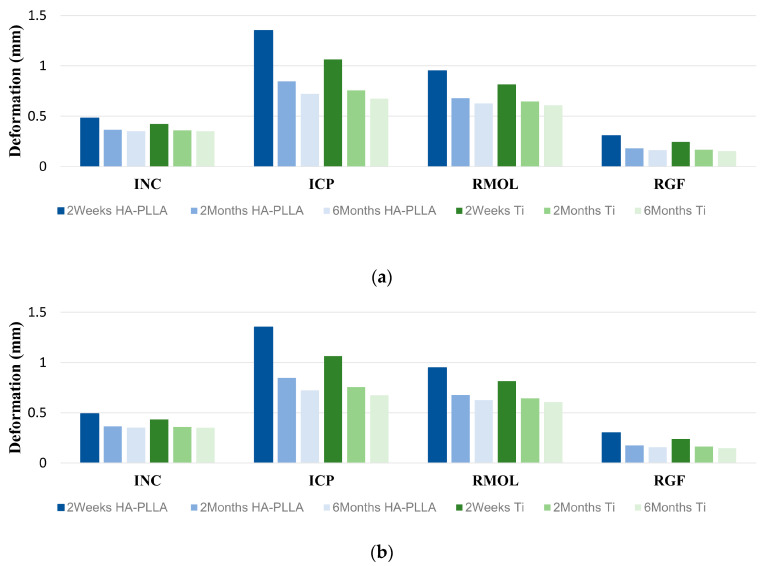
Deformation in the two types of fixation systems under functional masticatory loadings at 2 weeks, 2 months, and 6 months after surgery. (**a**) Miniplates and (**b**) screws.

**Table 1 bioengineering-11-01009-t001:** Material properties.

Components		HU	P_app_	Young’s Modulus (MPa)	Poisson’s Ratio
Titanium [28]		*-*	-	96,000	0.36
HA-PLLA [29]		-	-	9701	0.317
Cortical bone [30,31]	Healthy	-	-	15,000	0.33
Cancellous bone [30,31]	Healthy	-	-	1500	0.3
Tooth [32]		-	-	20,000	0.3
PDL [33]		-	-	0.69	0.45
Cortical bone [27]	2 Weeks	1399	1478.8	9238.29	0.33
	2 Months	1429	1514.8	9638.23	0.33
	6 Months	1534	1640.8	9854.52	0.33
Callus [27]	2 Weeks	212	54.4	27.44	0.33
	2 Months	308	169.6	203.45	0.33
	6 Months	376	251.2	406.49	0.33

**Table 2 bioengineering-11-01009-t002:** Nodes and element settings of mesh model.

Component	Number of Nodes	Number of Elements	Mesh Size	
			Maximum	Minimum
Miniplate	143,279	680,029	0.15	0.05
Screw	187,845	961,484	0.15	0.05
Cortical bone	377,281	1,853,115	0.8	0.15
Cancellous bone	357,839	1,909,969	0.8	0.15
Tooth	275,622	1,492,811	0.3	0.15
PDL	116,317	427,042	0.1	0.03
Fibular bone	88,182	423,717	0.8	0.15
Callus	13,097	63,035	0.8	0.15

**Table 3 bioengineering-11-01009-t003:** PVMS distribution across the fixations system.

	2 Weeks after Surgery				2 Months after Surgery				6 Months after Surgery			
Loading Conditions	Ti		HA-PLLA		Ti		HA-PLLA		Ti		HA-PLLA	
	Screw	Plate	Screw	Plate	Screw	Plate	Screw	Plate	Screw	Plate	Screw	Plate
INC	136.7	188.7	64.2	71.6	106.6	169.5	37.36	41.8	86.62	137.8	26.1	30.6
ICP	693.6	1005	198.7	268.0	379.3	549.9	84.44	112.8	275.9	397.4	59.7	73.8
RMOL	334.9	490.6	104.5	141.9	158.1	103.0	44.4	56.0	118.5	192.5	28.1	37.9
RGF	144.5	209.5	65.7	74.7	103.0	163.7	33.4	37.7	85.76	127.8	22.7	26.4

**Table 4 bioengineering-11-01009-t004:** Deformation measurement for the fixation systems.

	2 Weeks after Surgery				2 Months after Surgery				6 Months after Surgery			
Loading Conditions	Ti		HA-PLLA		Ti		HA-PLLA		Ti		HA-PLLA	
	Screw	Plate	Screw	Plate	Screw	Plate	Screw	Plate	Screw	Plate	Screw	Plate
INC	0.433	0.424	0.491	0.482	0.358	0.358	0.366	0.366	0.350	0.349	0.352	0.351
ICP	1.065	1.065	1.353	1.351	0.756	0.756	0.847	0.845	0.675	0.674	0.723	0.721
RMOL	0.814	0.818	0.948	0.952	0.644	0.644	0.677	0.678	0.607	0.609	0.626	0.627
RGF	0.239	0.245	0.301	0.307	0.163	0.163	0.176	0.182	0.148	0.153	0.157	0.162

## Data Availability

The datasets generated during and/or analyzed during the current study are available from the corresponding author upon reasonable request.

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
