# Peer review of "Biomechanical Evaluation of Hydroxyapatite/poly-l-lactide Fixation in Mandibular Body Reconstruction with Fibula Free Flap: A Finite Element Analysis Incorporating Material Properties and Masticatory Function Evaluation"

_bioengineering, 2024, doi:10.3390/bioengineering11101009_

Round 1

Reviewer 1 Report

Comments and Suggestions for Authors 1. For Table 1, please list references for each component. 2. Line 143, please clarify in more detail why cancellous bone was not a significant factor. 3. Is there mesh sensitivity analysis conducted to verify that results based on Table 2 are convergent? 4. Are there any experimental measurements available to verify the simulation? 5. Please expand the discussion on the contrast identified on line 295.

Reviewer 2 Report

Comments and Suggestions for Authors

The authors described "Biomechanical evaluation of hydroxyapatite/poly-l-lactide fixation in mandibular body reconstruction with fibula free flap: A finite element analysis incorporating material properties and masticatory function evaluation" through finite element analysis (FEA). The biomechanical integrity of Ti and HA-PLLA fixation systems used in reconstruction surgery with fibula flap was compared using FEA simulation. This study should be informative and attractive for potential readers. I have some question and suggestions to improve this manuscript.

1. Why did you choose HA-PLLA among biodegradable materials? As there are a lot of biodegradable materials in the world, they should add the reason in Introduction or Discussion.

2. How does this HA-PLLA compare with the One-plate fixation? The fibula is often thin, forcing fixation with a single large plate (Yamakawa S, et al. BMC surgery 2021). Please add the content of the one-plate fixation method with fibula flap in Discussion.

Round 2

Reviewer 2 Report

Comments and Suggestions for Authors

The authors revised the manuscript precisely. Thank you for this opportunity.